# Water Stress Index Detection Using a Low-Cost Infrared Sensor and Excess Green Image Processing

**DOI:** 10.3390/s23031318

**Published:** 2023-01-24

**Authors:** Rodrigo Leme de Paulo, Angel Pontin Garcia, Claudio Kiyoshi Umezu, Antonio Pires de Camargo, Fabrício Theodoro Soares, Daniel Albiero

**Affiliations:** School of Agricultural Engineering, University of Campinas, Campinas 13083-875, Brazil

**Keywords:** water stress, precision irrigation, non-water-stressed baseline, soil moisture, infra-red sensor

## Abstract

Precision Irrigation (PI) is a promising technique for monitoring and controlling water use that allows for meeting crop water requirements based on site-specific data. However, implementing the PI needs precise data on water evapotranspiration. The detection and monitoring of crop water stress can be achieved by several methods, one of the most interesting being the use of infra-red (IR) thermometry combined with the estimate of the Crop Water Stress Index (CWSI). However, conventional IR equipment is expensive, so the objective of this paper is to present the development of a new low-cost water stress detection system using TL indices obtained by crossing the responses of infrared sensors with image processing. The results demonstrated that it is possible to use low-cost IR sensors with a directional Field of Vision (FoV) to measure plant temperature, generate thermal maps, and identify water stress conditions. The Leaf Temperature Maps, generated by the IR sensor readings of the plant segmentation in the RGB image, were validated by thermal images. Furthermore, the estimated CWSI is consistent with the literature results.

## 1. Introduction

Precision Irrigation (PI) is a promising technique for monitoring and controlling water use that allows meeting crop water requirements based on site-specific data. Systems that use less water and have high energy efficiency are a reality. In recent years, through the development of technology in this sector, it has been possible to make irrigation systems that use PI more competitive, facilitating access for farmers. The detection of water stress is vital in PI management to ensure that plants are not subjected to stress levels that excessively restrict productivity or even production quality [1,2,3]. Detecting water stress is one of the critical problems in PI management to ensure that the plants do not experience stress levels that harm productivity or even the quality of production. For some crops, such as olive trees and viticulture, a certain degree of water stress causes the plant to produce more polyphenols and the acidity of its products to be smaller, benefiting its qualities.

Among the ways to detect water stress, methods that measure soil water tension, stomatal conductance potential, and sap flow stand out [4,5]. Soil water tension indicates the energy with which water is retained in the soil matrix and is, therefore, an indirect measure of the water stress to which the plant is subjected [6].

Under water stress conditions, there is stomatal closure, reduced transpiration, and a consequent increase in leaf temperature (T_L_) concerning ambient temperature. The Crop Water Stress Index (CWSI) enables quantifying the stress level by normalizing leaf temperature between the maximum (when there is no transpiration) and minimum plant temperatures (when transpiration is at its maximum potential).

Measuring the difference between a canopy and ambient temperature is an option for estimating plants’ water deficit levels. In plants under water deficit and subject to transpiration restrictions, an increase in the temperature of the leaf surface is expected since the vaporization of water on the surface of the leaves contributes to the cooling of these surfaces. Under these conditions, T_L_ approaches the ambient temperature (T_A_). On the other hand, when there are no water restrictions and transpiration is adequate, the leaf surface cooling results in different values of T_L_ and T_A_.

Kacira [6] detected water stress in flowers, identifying a faster response of this method than conventional methods, showing the advantage of using this index. For diagnosing water deficit, the calculated CWSI value can be compared with the critical CWSI value, which varies depending on the crop and growing stages. Usually, irrigation should be performed when CWSI is higher than 0.24 [7]. A CWSI value equal to or less than 0.2 can be used as a reference for plants without water stress [6].

The CWSI indicates crop water deficit and it can be useful to identify the moment to perform irrigations based on canopy and ambient temperature determinations. The CWSI value varies between 0 and 1, with zero indicating a plant without water restrictions. In contrast, a plant in conditions of extreme water deficit (without transpiration) has a CWSI equal to 1. In a simplified way, the CWSI can be calculated based on the difference between leaf and ambient temperatures, as shown in Equation (1).
(1)$$\mathrm{CWSI}=\frac{(T_{L}-T_{A})-{\left(T_{L}-T_{A}\right)}_{LL}}{{\left(T_{L}-T_{A}\right)}_{UL}-(T_{L}-T_{A})}$$ where (T_L_ − T_A_)_LL_ e (T_L_ − T_A_)_UL_ are the lower and upper limits, found by the Non-Water-Stressed-Baseline equation (NWSB). It is specific for each crop and correlates the difference between leaf temperatures and the environment with the Vapor Pressure Deficit (VPD) [8].

The literature on the CWSI is relatively vast. Studies focus on ways to obtain the index [9,10], in evaluating the behavior of crops concerning the CWSI [11,12,13,14], and in the ways to use the information in irrigation management systems [15,16,17]. All these studies have in common the use of similar methodologies for data acquisition, such as the use of infrared (IR) temperature sensors and thermal cameras to obtain the leaf temperatures of plants. However, in small-scale applications, the high cost of thermal cameras can be a limiting factor for getting thermal maps, and IR sensors can be a technically viable and lower-cost alternative.

Factors such as the high cost of acquisition, dependence on the international exchange rate, the difficulty of operation, and sensitivity to calibration are obstacles to using thermal cameras to obtain a thermal map. However, IR sensors are more accessible and simpler. This paper hypothesizes that developing a device made up of low-cost IR sensors combined through an algorithm for image processing can be used to identify plants’ water stress.

Therefore, the general objective of this work is to develop a water stress detection system using T_L_ indices obtained by crossing the responses of infrared sensors with image processing.

## 2. Material and Methods

Indoor experiments were performed at the Faculty of Agricultural Engineering (Feagri/Unicamp, Campinas, SP, Brazil), located at 22°48′57″ S, 47°03′33″ W, average altitude of 640 m. The climate zone is classified as humid subtropical climate (Cwa according to the Koeppen classification), with a decrease in winter rainfall and an annual average temperature of 20.7 °C. Winters are dry and mild, while summers are rainy with moderately high temperatures. The hottest month of the year (February) has an average temperature of 23.4 °C, and the coldest (July) has an average of 17.2 °C [18]. The experiments were performed from 25 May 2020 to 28 June 2020, including one crop cycle.

The study area consisted of an indoor test bench of 0.75 m × 1.50 m × 0.25 m, where arugula (*Eruca sativa miller*) was cultivated in the spacing of 0.20 m between rows and 0.05 m between plants [19]. Soil was used as a growing medium for the plants. Arugula was chosen for the experiments because of its ease of cultivation and homogeneous distribution of leaf shape. Figure 1 illustrates steps of the adopted methodology: (i) acquisition of temperature, relative humidity, and plant height data; (ii) image acquisition and processing; (iii) reference thermal image acquisition; (iv) parameterization of the NWSB equation; and (v) soil moisture monitoring.

### 2.1. Instrumentation

The bench was instrumented with infrared, ultrasonic, temperature, and humidity sensors and an optical camera. Figure 2 shows the system’s configuration in which the tests were performed, with the sensors and embedded devices for data processing.

Plant height monitoring was performed using a digital ultrasonic sensor model HC-SR04, with an operating range between 2 mm and 4 m and ±3 mm precision. Literature mentions that the distance between the temperature sensor and the leaf surface influences the measurement quality. Therefore, the vertical distance adopted was 50 mm to improve the quality of temperature determinations in this project, considering the small size of the arugula. The adopted distance was also used for the crops of lettuce [20], saffron [21], and black pepper [12], and the results obtained by these studies showed that temperature readings at this distance were adequate for monitoring plant water stress.

This experiment used an IR temperature sensor (MLX90614-ESF-BAA, Melexis, Belgium), with an operating range between −70 e 180 °C, ±0.5 °C precision, and 90° conical field of view (FoV), with the emissivity of the adjustable sensor between 0 and 1.00. In the experiments, the equipment was regulated to capture the emissivity of vegetable leaves equal to 0.98 [22,23]. The temperature measured by the IR sensor, also called the object temperature (T_o_), is the mean of all temperatures within the reading area. As the FoV is conical, its reduction will decrease the reading area and, therefore, it will present a more accurate reading, since both are directly proportional. A metallic apparatus was used to reduce the FoV in the sensor that collimates the infrared beam. Since the IR sensor reading area is a function of the height of the set of the sensor set, this must be kept constant [24].

A sensor (AM2302, Aosong Electronic Co., China) was used to measure ambient temperature and humidity in an operating range between −40 to 80 °C and 0 to 100% RH, precision ±0.5 °C, and ±2%RH. This sensor is widely used for its precision and ease of communication with embedded devices [25,26,27].

The digital camera of a smartphone (iPhone 7—A1778, Apple Inc., United States) was used to obtain the RGB images with the specification: 1/3” sensor, 12 MP resolution, f/1.8 aperture, 28 mm equivalent focal length. Capacitive sensors (EC-5, Decagon Devices Inc., United States) were used to monitor soil moisture, with a measuring range from 0 to 0.60 m^3^ m^−3^, a precision of 0.02 m^3^ m^−3^.

### 2.2. Soil/Plant Thermal Map

The bench test uses stepper motors to move the set of sensors three-dimensionally along the X, Y, and Z axes. Using local coordinates, the set of devices ran through the entire test area, performing a daily pre-scan with a fixed height of 0.5 m above ground level to determine the plants’ height range through the ultrasonic sensor. Then, according to the maximum size obtained, the height of the set was adjusted so that the infrared sensor reading was 50 mm above the plants’ level. The distance of the set was measured with the aid of a measuring tape.

A daily scan was performed, taking static readings of information from IR sensors and ambient temperature and humidity throughout the growing period, always at 12:00 pm. This time presented the best correlation between CWSI and leaf water potential (Y_L_) [28]. A pixel with a size of 20 × 20 mm was established where, at each point, the X, Y coordinates and the T_O_ values were known, in addition to the T_A_ and RH at the time of reading. It is worth mentioning that the information was read in places with plant and soil, only plant, or only soil, according to their location. Therefore, it was possible to generate a Soil/Plant Thermal Map of the experimental area (60 × 21 pixels) with the T_O_ information. At the same time, the other two parameters read were used to calculate the CWSI.

### 2.3. Image Processing

Computer vision techniques were applied for the identification and segmentation of leaves and soil.

The tests were carried out with the RGB camera and a desktop computer (Vostro, Dell Inc., Brazil) with a 3.90 GHz, 8th generation Intel Core i5 processor, 12 GB of RAM and a 2 GB NVIDIA GeForce MX 130 video card and Windows operational system (Windows 10 Pro, Microsoft Inc., Redmond, Washington, United States).

The algorithm for image processing was built in *Python* language and used the following libraries: *Open-CV* for commands related to image manipulations; *Numpy* for the realization of the mathematical operations involved in the process; and *Pandas* for grouping and organizing the output data. The ExG (RGB) model was used for image segmentation [29], previously evaluated as the model that presented the best performance indices under the conditions of the experiment.

During the cultivation cycle, 15 images were taken in the same position perpendicular to the bench. Segmentation quality was quantified by the F-score [30,31]. The F-score is defined as the harmonic mean between precision (number of correct positive classifications concerning the total positive classifications) and sensitivity (number of correct positive classifications about pixels belonging to plants). This metric quantifies the classification of images robustly between the absence of True Positives (TP) as 0 and a perfect classification as 1. In addition, the data were also evaluated using descriptive statistics.

The images that served as a reference to determine whether the classification was correct or not were manually classified. The binarization of each image was done through the mean of 10 pixels selected using the RGB values of the plants. Pixels within the range of values between mean, plus or minus one standard deviations, were considered plant (value 1) so that pixels containing soil received a value of 0.

### 2.4. Leaf Temperature Map Obtaining

The difference between leaf (T_L_) and ambient (T_A_) temperatures is one of the parameters used in the CWSI equation. The Leaf Temperature Map (LTM) was obtained by crossing the image pixel by pixel of segmented plants [30] and the soil/plant thermal map, where only the plants were considered. Thus, the LTM was obtained by applying the generated mask on the thermal map, preserving only plant temperatures.

The reference temperature of surfaces was determined according to [23]. To estimate the temperature of the wet reference (T_MIN_), three plant leaves were sprayed with water on both sides, simulating the fully transpiring condition, approximately 1 min before imaging acquisition. In addition, another three leaves, on the same day, were covered with petroleum jelly, simulating non-transpiring leaf condition (T_UL_) [32].

### 2.5. Leaf Temperature Map Validation

The LTM validation was made by comparing it with the image generated by the IR camera, positioned perpendicular to the ground. The thermal images obtained by the IR camera were processed by the manufacturer’s software. The validation was done by analyzing, pixel by pixel, the difference between the temperatures read by the IR sensor (S_x,y_) and by the camera (C_x,y_), as shown in Equation (2) and the general mean error calculated by Equation (3), where *n* is the number of pixels of the generated maps. In addition, the mean standard error was also calculated.
(2)$$Error{\;}_{x,y}=\frac{S_{x,y}-C_{x,y}}{C_{x,y}}$$
(3)$$FGeneral\;mean\;error=\frac{\sum error_{x,y}}{n}$$

### 2.6. Parameterization of the Non-Water-Stressed Baseline

The NWSB equation was parameterized with data obtained from a second experimental bench, with characteristics equal to the main bench, such as the same dimensions, soil type, quantity, and plant spacing. In addition, the two benches were in the same environment, enabling the sharing of data and conditions.

A second AM2302 sensor was used to take the T_A_ and RH readings, and another IR sensor, arranged at fixed points to read the T_L_ of the plants. It is essential to highlight that as this sensor did not perform a scan but fixed readings, it was positioned so that only the arugula leaves were within the sensor reading area [28].

Measurements were taken with an interval of 10 min for 30 days, corresponding to the arugula cultivation cycle. Therefore, in addition to considering two planting cycles for the parameterization of the equation, it was necessary to calculate the *VPD* through the T_A_ e RH values read by the AM2302 sensor (Equation (4)).
(4)$$VPD=\left(1-\mathrm{RH}\right)\times 0.6108\times 10^{\frac{7.5T_{A}}{273.3+T_{A}}}$$

By correlating the values of (T_L_ − T_A_) and VPD, the NWSB equation corresponds to the linearized model of the set of generated points. Therefore, this equation was used to calculate the water stress of plants on the principal test bench. Following Testi [33] methodology, (T_L_ − T_A_) and the VPD could be correlated according to the time of measurements, as they vary throughout the day. However, we chose to use a simplified correlation and only a daily mean value for this study since scanning the points and calculating the CWSI were performed only once a day. The curves obtained by these correlations provided the NWSB equations.

The lower limit (T_L_ − T_A_)_LL_ of the CWSI was calculated using the NWSB equation, with the VPD values of the environment at the time of reading during the daily scan of the sensors, which can be understood as the linear coefficient of the equation obtained. The upper limit (T_L_ − T_A_)_UL_ calculation followed the methodology used by [34]. This methodology uses a potential VPD to simulate the increase of (T_L_ − T_A_) when the VPD is zero at ambient temperature, generating a new value of VPD that must be used in the NWSB equation, resulting in the maximum limit of the temperature difference.

### 2.7. Crop Water Stress Index Calculation

It was possible to calculate the CWSI by Equation (1) with all parameters defined, quantitatively finding the degree of stress of the plant at the moment of the scan.

The water stress map generated by the CWSI was validated by making comparisons with soil moisture information obtained with EC-5 sensors (Figure 3). The experimental bench was separated into three treatments, each subjected to its irrigation regime, with different water conditions.

Soil moisture close to the condition of Field Capacity (FC) was determined before seedling transplanting, saturating the local soil, and monitoring the moisture until it reached a steady value. From the FC information, water stress levels were differentiated in 3 treatments in which irrigation was performed whenever the moisture got a critical value corresponding to T_1_ = 85% FC; T_2_ = 75% FC; T_3_ = 55% FC. The volume of water applied in each treatment area and each irrigation event was fixed (0.7 L) because the automatic system reservoir has this volume. Thus, for T_2_ and T_3_ were conditioned continuous levels of water stress for the plants.

The sensor reading routine considered 20 min for each measurement and, thus, the system irrigated each treatment individually when the sensors indicated critical moisture corresponding to each treatment.

As they do not present a regular pattern, both the CWSI and the soil moisture were statistically evaluated by the Exponentially Weighted Moving Average (EWMA) chart [35]. EWMA averaging is a statistical quality engineering method in which data are depicted on a control chart and presented against an exponential moving average immune to data normality issues [36]. This method makes it possible to assess whether the process is stable and capable (within the control limits).

### 2.8. Leaf Temperature Map Validation

With the results of the generated thermal map and the image segmentation, it was possible for the crossing of information and the overlap of these data to generate an LTM (Figure 4). For this, pixel-by-pixel multiplication of points on the temperature map by values 0 or 1 occurred, according to the processed image.

The resizing of the images was essential for overlaying the maps since the RGB image originally had 5000 × 2700 pixels, and the generated temperature map had 61 × 22 pixels, so the final dimensions of the LTM are 1280 × 620 pixels. It can be observed that the treatments presented different ranges of leaf temperatures, where treatment 3, which was irrigated only when the Volumetric Water Content (VWC) reached 55% of FC, had the highest T_L_ compared to treatments 1 and 2, irrigated in 85% and 75% of FC, respectively. This behavior is expected since the water conditions of each of the treatments are different, causing less water to be available for the plants and, consequently, different levels of stomatal closure.

### 2.9. Non-Water-Stressed Baseline Equation

Values of temperature and relative humidity were measure every 10 min and the corresponding values of VPD were calculated. Figure 5 shows the monitoring of T_O_, T_A_, and RH.

Leaf temperature values were consistently below ambient temperatures, which satisfies the equation’s parameterization condition, which only foresees plants without water stress. The correlation between temperature difference and VPD resulted in the equation (Figure 6).

An equation was also parameterized considering an unprotected environment for evaluation purposes, whose monitoring of ambient and leaf temperatures and RH is shown in Figure 7.

According to Montgomery [36], a linear regression must be validated from a residual analysis, where this analysis must show that the residual data between the variables do not have bias and have constant variance. The behavior of the parameterized equations for the two environments was adequate since there is no bias between the variables and the mean of the residuals was equal to zero [37].

Although the angular coefficients of NWSB_D_ are in the range of values close to those found in the literature for other crops (Table 1), it was observed that the calculated limits (T_L_ − T_A_)_LL_ and (T_L_ − T_A_)_UL_ did not represent the real relationship between the difference in leaf and ambient temperatures and the VPD, since the experimental bench was in a protected cultivation environment, without the presence of factors such as wind, dew, and large variations in temperature. The NWSB_P_ coefficients were shown to be smaller than those found in parameterizations of other crops.

Berni et al. [39] obtained as angular and linear coefficients the values −0.35 and 2.08, respectively. They concluded that a lower angular coefficient is observed in plants with a more dependent relationship with the atmosphere and climate, as is the case with olive trees, which have a low temperature variation for significant variations in the VPD [40]. Furthermore, the authors pointed out the small size of the leaves concerning the tree canopy and even that, for several crops, a certain level of stomatal closure is common when there is an increase in the plant transpiration.

The parameterization of the NWSB must occur under the same climatic conditions in which the CWSI will be calculated [41]. The parameterization must happen that way so that the normalization of temperatures does not occur in a displaced way, causing the lower and upper limits of (T_L_ − T_A_) to be outside the actual range of temperature variation.

Idso [7] and Xavier [5] also stated that the shape and size of the leaves could affect the angular and linear coefficients of the NWSB, something relevant in this work since the data were collected every 10 min throughout the entire crop cycle. Due to different water requirements, the plant development stage also affects these coefficients [7]. In the literature, some studies collected data for 2 years in pistachio crops [33]. Still, in addition to being large plants, their productive age can reach 25 years [42]; that is, the leaves physical variations were not significant to affect the NWSB regression. Other works parameterized the equation with data collected for only 3 days, at specific times (8:30 a.m.; 11:30 a.m.; 2:30 p.m.; 5:30 p.m.; 8:00 p.m.) [9], and it was observed that temperature variations in times before 9:00 a.m. and after 4:00 p.m. do not show good relations with the VPD.

The low slope of the NWSB_P_ straight line and the dispersion of points show that arugula can be a crop that presents a small range of differences between T_L_ and T_A_ concerning the environment, corroborating the results found by Berni et al. [39]. As a result, the CWSI may be more sensitive to temperature variations since the normalization of temperatures can be affected by the maximum and minimum limits found by NWSB.

## 3. Results and Discussion

The validation of the LTM was done through the image obtained by the thermal camera (Figure 8), where it is possible to identify that the minimum temperatures were 19.2 °C. The generated image exceeds the limits of the experimental area, causing the upper range of temperature values to be outside the maximum temperature range of the crop. The plant segmentation image was also superimposed on the thermal image to eliminate the part that does not correspond to the area of interest. Only pixels with plants were evaluated.

The mean error between temperatures measured by the IR sensor and the camera was −0.1959 °C; that is, the temperatures measured by the sensor were, on the mean, 0.2 °C below the temperature measured by the thermal camera. The standard error was equal to 0.03%. This sensor is widely used in agricultural studies due to its low cost and the good results presented in the literature [43,44,45,46]. Despite the 0.2 °C mean error found being within the precision provided by the manufacturer of 0.5 °C, it can be observed that the scale of values obtained by the thermal camera has points with up to 2.5 °C of sensor difference. This difference can be explained by the resizing of maps and images.

These errors are similar to those obtained by Osroosh [47], who revealed that the primary source of error was due to slight misalignment of the RGB and thermal images leading to the inclusion of background in the final masked thermal image. Gimenez-Gallego [48] also obtained similar results. Another error, identified by Zhou [49], is due to the low contrast of the thermal image; this problem, according to Zhou, could be addressed by calibration using a thermally controlled flat plate blackbody. However, the interpolation is done in a bilinearized way and by the difference in the number of original pixels. Therefore, the value of the resized pixels may have been changed.

Irrigation events were recorded to be related to soil moisture data. Monitoring soil moisture (Figure 9) throughout the crop cycle shows that Treatment 1 had the highest water availability in the soil for the plants, with its WVC (Volumetric Water Content) varying between 85% and 97% of the field capacity of the medium used. Treatment 2 had moisture values ranging between 75% and 84% of FC, and Treatment 3, in turn, had its WVC between 50% and 70%. Figure 9 shows that the behavior of the parameterized equations for the two environments was adequate since there is no bias between the variables and the mean of the residuals was equal to zero [37].

### 3.1. Leaf Temperature Map

The scans performed generated points with local coordinates and the values of T_O_, T_A_, and RH. However, these last two are only used to calculate the CWSI. Figure 10 shows the map generated on the 25th Day After Planting (DAP), with a complete set of points. Since the scans were daily, a thermal map per day was obtained.

It is possible to notice that the observed temperatures vary between 19.5 °C and 22.5 °C, results consistent with similar experiments carried out by Osroosh [47], Gimenez-Galego [48], Parihar [50], and Zhou [49] that presented amplitude-absolute values close to our experiment and errors within a range similar to our errors, which demonstrates that the use of this low-cost apparatus has adequate capacity to be helpful in irrigation management. Furthermore, Parihar [50] presents exciting results and states that their results support the increased use of on-the-ground thermal imagery to schedule irrigation in horticulture plants. Still, it is essential to emphasize that points without any plant segmentation were read in this map: pixel readings containing only soil, only plant, and the junction of the two.

Points with only soil presented higher temperatures than the ones with soil and plant or just plant. This problem was reported by Osroosh [47], who suggested the use of algorithms to overcome this challenge. Following this recommendation, we use image processing methods so that only the leaf temperatures must be included for the CWSI calculation to be valid; therefore, the image processing step was performed in the sequence.

### 3.2. Image Processing

The 15 images obtained throughout the crop cycle were processed, and a new output image was generated for each of them; that is, a binary image with the segmentation of plants. As the images were captured at 5, 15 and 25 days after planting (DAP) (Figure 11), it was possible to observe different shapes and sizes of the plants and, therefore, to evaluate the robustness of the method used.

In addition to the difference between plant density in the experimental area throughout crop development, differences in soil color caused by its moisture are factors that can generate disturbances and affect the results generated in image processing. It is noteworthy that, for this step, there was no division between treatments. By manually classifying the images as a reference in the evaluation of the model, segmented images were generated (Figure 12a). Subsequently, the pixels with plants received a value of 1 (Figure 12b). Finally, applying the ExG (RGB) equations, it was possible to obtain the model metrics (Table 2), considering the 15 imagens as repetitions.

The F-score was greater than 85%, and the accuracy was close to 90%, indicating an adequate performance of the image segmentation method. The average processing time was 1.5 s, indicating a low computational cost. Although this method has relatively low sensitivity values, its precision–that is, the number of pixels classified as plants that were plants–made the total error of the method less than 10%. The performance of this method corroborated the results of Perissini [30], who also indicated the ExG (RGB) with the best *F-score values* in their evaluations. However, the processing time found by the author was higher than in this study. This difference is due to the different environmental conditions when the images were captured and shows the complexity of image processing, since similar techniques and approaches can produce different results.

### 3.3. Crop Water Stress Index

By calculating the CWSI using the leaf temperature values obtained from the LTM, T_A_, and RH obtained by the AM2302 sensor and the limits of the NWSB_P_ parameterized for the protected environment conditions, it was possible to generate a water stress map (Figure 13) for the treatments of the experimental area.

As expected, the behavior of the crop in relation to the environment affected the limits (T_L_ − T_A_)_UL_ and (T_L_ − T_A_)_LL_ obtained from the NWSB equation, causing the CWSI calculation to present values slightly higher than 1. This would indicate that the plants were under 100% water stress, a result which is consistent with similar experiments carried out by Luan [51], Parihar [50], and Katimbo [52], who carried out similar experiments about water stress and obtained values similar to ours, which indicates that the use of this new low-cost system meets specifications in terms of precision and accuracy to generate reliable data for irrigation management through CWSI. Still, these characteristics were not verified when visually evaluating the state of the leaves at the points that presented these values, showing that the temperature normalization range was displaced concerning the actual leaf temperatures of the crop. Some negative values were also found, indicating that the plants were not under any water stress; indirectly, this occurs when the canopy vapor transport resistance ($r_{a}$) is lower than the aerodynamic resistance of the air ($r_{c}$) [53]. This fact is also verified by Katimbo [52], who states that it is not surprising that some of the CWSI values were negative. This has also been observed in other studies that have used empirical bases to calculate CWSI, such as DeJonge [54]. Katimbo [52] explains that the CWSI tended to overestimate stress with high values; however, this can be due to a cooler lower baseline or uncertainties around the computation of aerodynamic resistance.

CWSI values less than 0 and greater than 1 are reported in the works of Barbosa [53], Hernández [55], and Quebrajo [56]. As the values obtained were close to the limit normalized by the index, a correction was made so that negative values received 0 as their new value and values greater than 1 were reduced to the CWSI limit. When calculating the CWSI using the limits obtained by the NWSB_D,_ it was possible to identify that the normalization of (T_L_ − T_A_) values was significantly displaced concerning the actual range of temperature variations. The CWSI map of the crop (Figure 14) generated for the unprotected environment showed high values compared to the map developed for the experimental bench environment. Plants from treatment 1 had CWSI values close to 0.8, while in the map in Figure 15, their values were almost all zero.

Just for evaluation purposes, the empirical method was also calculated at 25DAP to find the *T_UL_* and *T_LL_* limits, following Camoglu [12]. One of the leaves was wetted with water 1 min before the temperature reading, serving as a reference for a surface with maximum transpiration and revealing the minimum temperature that the plant would reach, 19.3 °C. The other leaf, covered with Vaseline about 30 min before the reading, served as a surface where there is no transpiration, indicating as 24.2 °C the maximum temperature that the plant could reach under the climatic conditions of the day, presenting T_A_ = 22.1 °C and RH = 47% at the time of measurement. The generated CWSI map (Figure 15) shows that the values did not exceed the limits between 0 and 1 of the index and that the maximum stress level read was 0.45, while the map generated through NWSB_P_ presented values equal to 1. At the same time, it can be observed that in treatment 1, the CWSI values were between 0.1 and 0.3, whereas in Figure 13, the majority of pixels identified as plants had values equal to 0.

Although the use of natural reference surfaces for the determination of temperature limits results in a range of CWSI values more minor than those presented by the protected environment equation, the use of this methodology makes it difficult to process automation because it requires surfaces for all scans that are made of the tillage area. Bellvert [28] calculated the CWSI and compared it with local methods of measuring leaf water potential ($\psi_{L})$ for viticulture, Çolak [57] for quinoa, Cohen [58] for cotton, and Mastrorilli [59] for soybean. Their results confirmed existing consensus in the literature that there is a direct relationship between $\psi_{L}$ and the CWSI. As shown in Table 3, it can be seen that for non-woody crops, there is a tendency for the CWSI to have a limit for a well-irrigated situation slightly higher (0.25) than that of woody crops (0.2), and the same trend can be observed for the leaf water potential. Furthermore, the authors concluded that the value of 0.25 should be used to define an irrigation regime based on CWSI for the arugula crop, which is consistent with the studies carried out by [60] for safflower, [59] for soybean, and [57] for quinoa.

When correlating the CWSI with the productivity in the eggplant crop, Çolak [11] also obtained the best results for index values equal to 0.2, noting the productivity fell from 78.7 to 40.9 t ha^−1^ when the CWSI increased to 0.6. As for the watermelon crop, the productivity of treatments with the indices 0.2, 0.4, and 0.6 showed no statistical differences, showing that the crop resists higher levels of water stress without harming its development. Therefore, an irrigation system based on a CWSI equal to 0.6 would require less frequent irrigation events.

At 25DAP, the date on which the maps presented in this work were generated, CWSI mean values were 0.0072, 0.2731, and 0.3840 for treatments 1, 2, and 3, respectively. Already the moisture values showed 91.21% FC, 79.50% FC, and 58.80% FC. Thus, throughout the entire crop cycle (Figure 16), it was possible to observe that the stress levels increased when the soil moisture of each treatment decreased, revealing that the proposed system could detect the water stress caused by different water conditions.

The above results indicate that one of a plant’s responses to reducing water availability is to increase leaf temperature caused by stomatal closure [61]. Thus, the CWSI is a good indicator of water stress. It can be used to define the irrigation management of individual plants or areas with the same treatment, corroborating the results obtained by Ben-Gal [13], Fattahi [16] and Ciezkowski [62], who defined irrigation management using a CWSI limit value for different crops. These values vary according to the stage of crop development [63], revealing the importance of monitoring stress levels throughout the entire plant cycle.

As discussed by Alchanatis [64], the use of this methodology requires the measurement of simple parameters such as plant and air temperatures and relative humidity, dispensing with the help of complex instruments or sensors. By analyzing the EWMA charts (Figure 17), it was possible to observe that none of the points selected throughout the crop cycle were outside the upper (LSC) and lower (LIC) control limits, indicating the stability of the process and corroborating the results obtained by Albiero [35] and Holt [65]. If any of the points were out of limits, the process should be considered unstable.

These results show the feasibility of this new system formed by a low-cost IR sensor combined with adequate image processing. This approach is also used by [47], who developed a thermal–RGB system that obtained interesting results in evapotranspiration analysis, obtaining real-time prescription maps, and irrigation scheduling.

## 4. Conclusions

It has been demonstrated that it is possible to use low-cost IR sensors with directional FoV to measure plant temperature, generate thermal maps, and identify water stress conditions. The Leaf Temperature Maps, generated by the IR sensor readings of the plant segmentation in the RGB image, were validated by thermal images.

The ExG (RGB) method, used for the segmentation of RGB images, showed good classification metrics, between 86% to 90% of correct classifications, and low computational cost, reducing the need for high-performance computers and enabling the use of embedded devices. The parameterization of NWSB showed that the plant responds in different ways in different climatic conditions.

The proposed system detected different CWSI values for soil moisture variations since the stress index increased with the reduction of water availability. The EWMA charts indicated the stability of the process. The results achieved in this work suggest future studies for the creation of a control system based on the CWSI to define the irrigation management in an automated irrigation system.

## Figures and Tables

**Figure 1 sensors-23-01318-f001:**
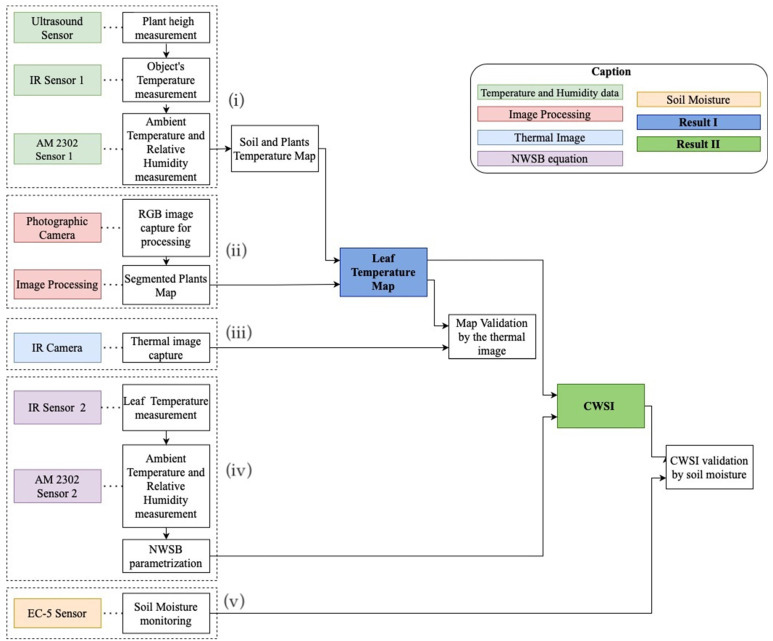
Steps of the methodology adopted for detection of water stress through the CWSI.

**Figure 2 sensors-23-01318-f002:**
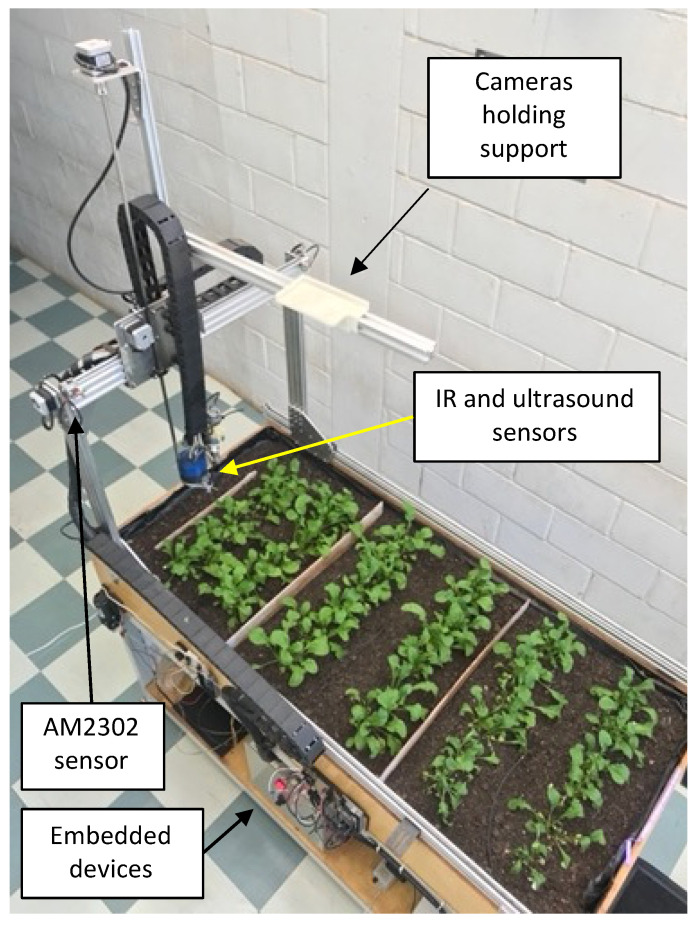
Schematic of the bench test, showing the arrangement of sets of sensors to detect water stress through CWSI.

**Figure 3 sensors-23-01318-f003:**
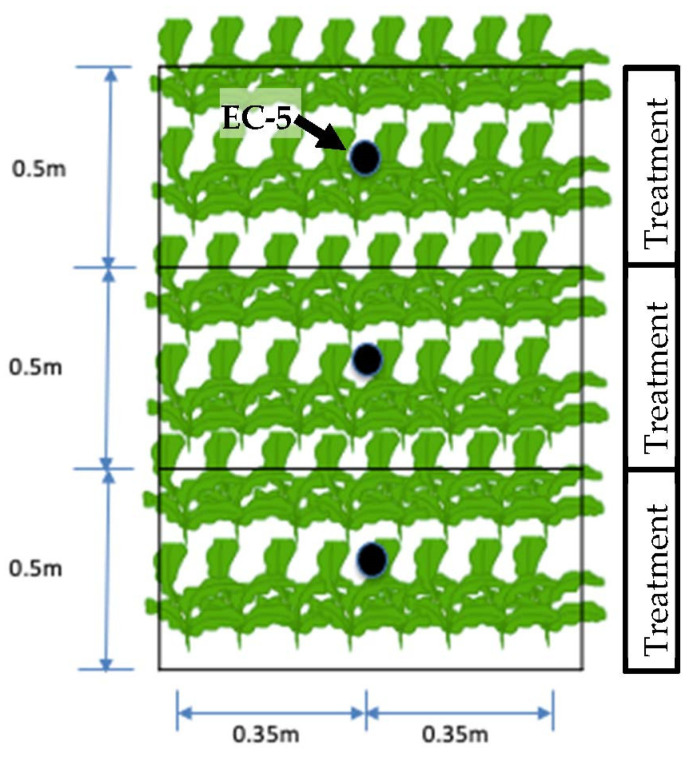
Distribution of soil moisture sensors placed in the center between the rows of each treatment in the experimental area.

**Figure 4 sensors-23-01318-f004:**
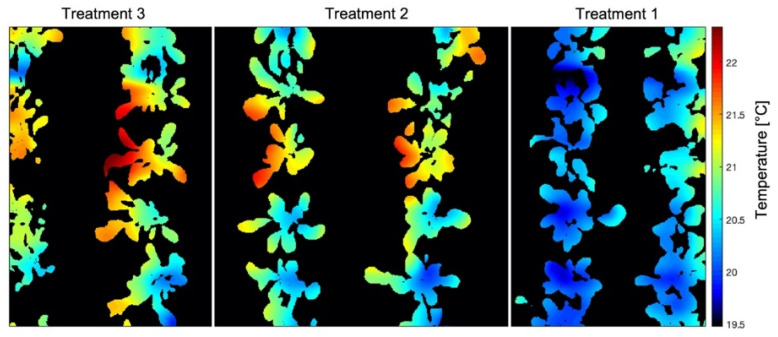
Leaf Temperature Map generated by crossing the temperature map of the experimental area with the segmented image of the crop—25DAP (Day After Planting). Treatment 1 (T_1_ = 85% FC (Field Capacity)); Treatment 2 (T_2_ = 75% FC); Treatment 3 (T_3_ = 55% FC).

**Figure 5 sensors-23-01318-f005:**
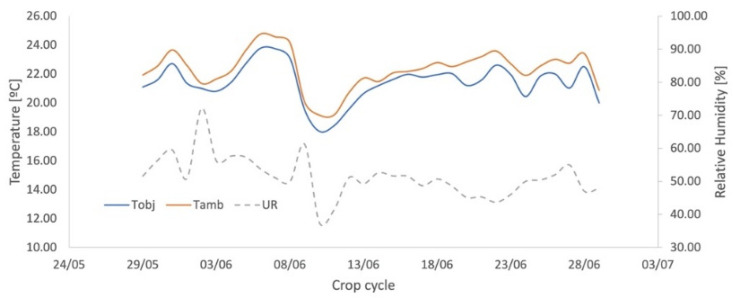
Ambient (T_amb_) and Object Temperature (T_obj_) monitoring and Relative Humidity (UR) for the auxiliary bench in NWSB parameterization.

**Figure 6 sensors-23-01318-f006:**
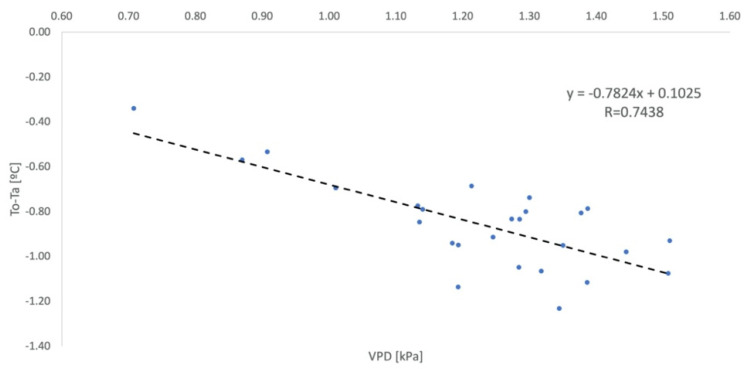
Non-Water-Stressed Baseline Parameterization defined for the protected environment (NWSB_P_). Vapor Pressure Deficit (VPD).

**Figure 7 sensors-23-01318-f007:**
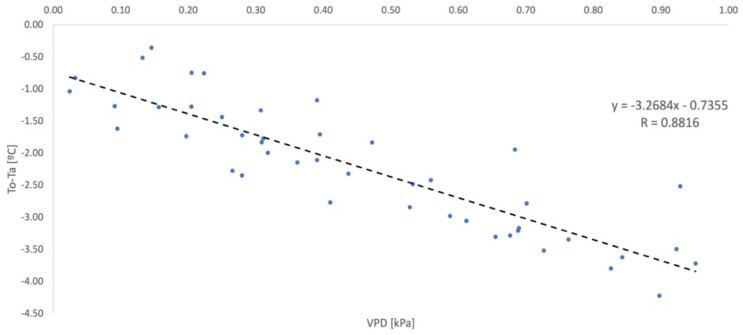
Non-Water-Stressed Baseline Parameterization defined for the unprotected environment (NWSB_D_). Vapor Pressure Deficit (VPD).

**Figure 8 sensors-23-01318-f008:**
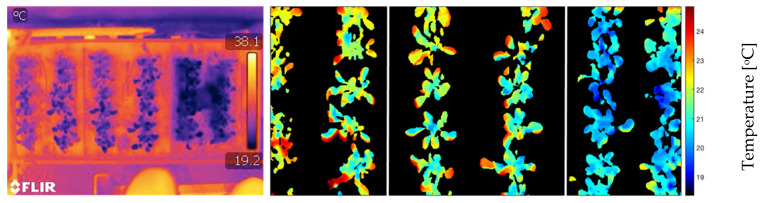
Generated thermal image, at 25DAP (Day After Planting), by the (**left**) original and (**right**) resized IR camera, with the overlay of the processed image for segmentation of the plants.

**Figure 9 sensors-23-01318-f009:**
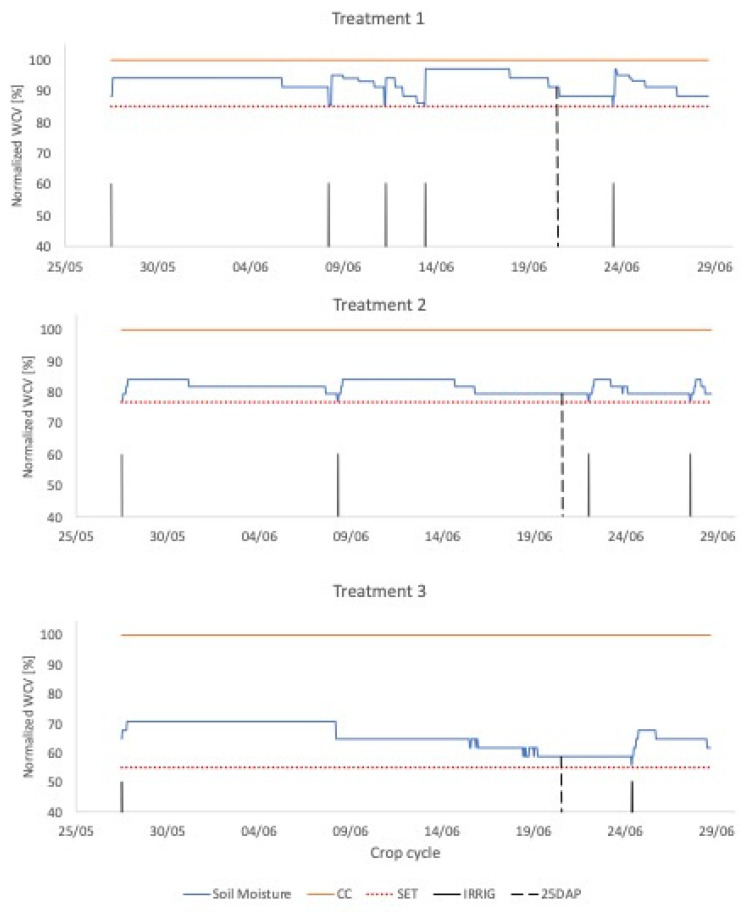
Moisture monitoring for each treatment throughout the crop cycle, where the VCW (Volumetric Water Content) values were normalized. Treatment 1 (T_1_ = 85% FC (Field Capacity)); Treatment 2 (T_2_ = 75% FC); Treatment 3 (T_3_ = 55% FC).

**Figure 10 sensors-23-01318-f010:**
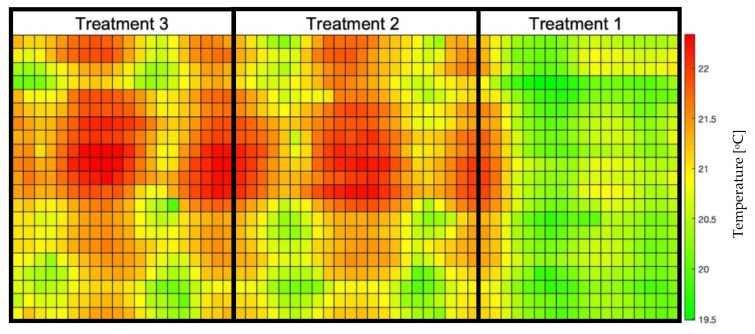
Thermal map of the experimental area, showing points of soil, plant, or a combination of both. The map was generated at 25DAP (Day After Planting), the final stage of culture. Treatment 1 (T_1_ = 85% FC (Field Capacity)); Treatment 2 (T_2_ = 75% FC); Treatment 3 (T_3_ = 55% FC).

**Figure 11 sensors-23-01318-f011:**
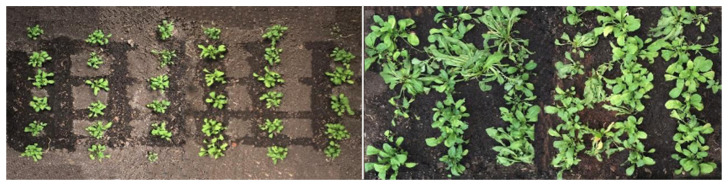
Images of the crop at different stages of development, used in the evaluation of processing methods.

**Figure 12 sensors-23-01318-f012:**
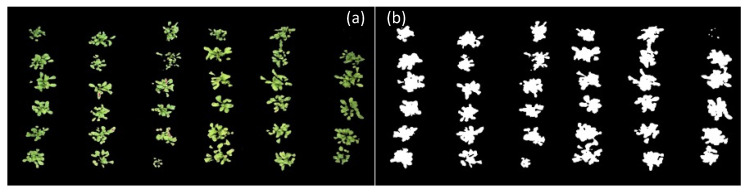
Images used as a reference for evaluating approaches, where (**a**) the pixels were segmented within the mean of the 10 selected pixels and (**b**) received a value equal to 1.

**Figure 13 sensors-23-01318-f013:**
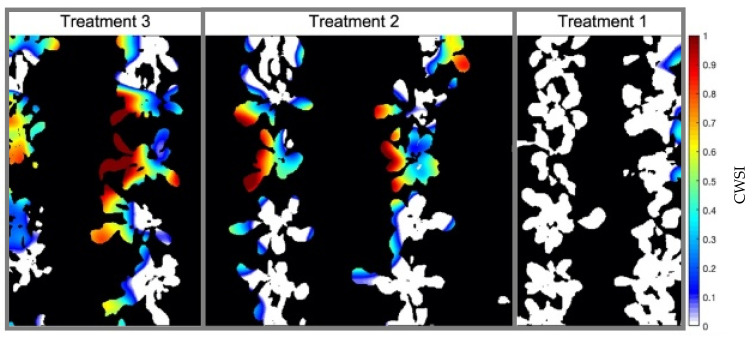
CWSI (Crop Water Stress Index) map of the experimental area in a protected environment generated for the 25DAP (Day After Planting), in a scan carried out at 12:00 p.m., the time with the best reading results. Treatment 1 (T_1_ = 85% FC (Field Capacity)); Treatment 2 (T_2_ = 75% FC); Treatment 3 (T_3_ = 55% FC).

**Figure 14 sensors-23-01318-f014:**
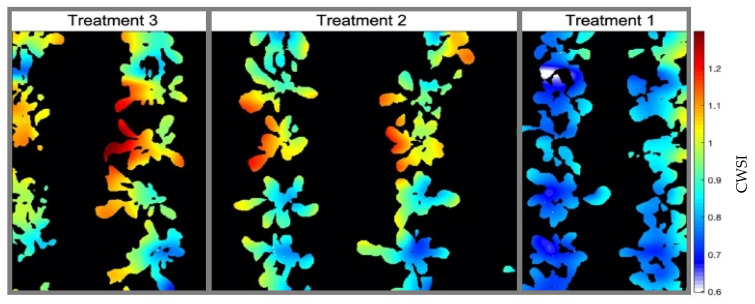
CWSI (Crop Water Stress Index) map of the experimental area in an unprotected environment generated for 25DAP (Day After Planting). The upper and lower limits of NWSB_D_ did not indicate actual temperature values, and the minimum calculated CWSI was 0.6, corresponding to the situation without water stress. Treatment 1 (T_1_ = 85% FC (Field Capacity)); Treatment 2 (T_2_ = 75% FC); Treatment 3 (T_3_ = 55% FC).

**Figure 15 sensors-23-01318-f015:**
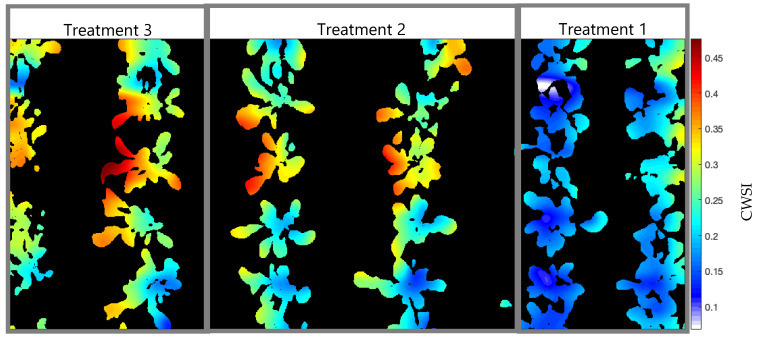
CWSI (Crop Water Stress Index) map generated for 25DAP (Day After Planting), using the empirical method to obtain the T_UL_ (Temperature Upper Limit) and T_LLN_ (Temperature Lower Limit) through natural reference surfaces. One of the leaves was wetted with water to simulate 100% transpiration and the other was covered with Vaseline to block leaf transpiration. Treatment 1 (T_1_ = 85% FC (Field Capacity)); Treatment 2 (T_2_ = 75% FC); Treatment 3 (T_3_ = 55% FC).

**Figure 16 sensors-23-01318-f016:**
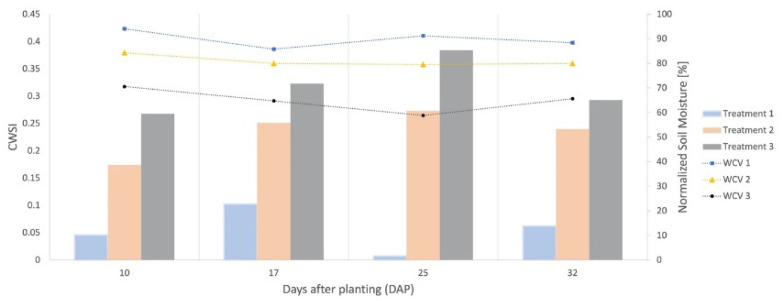
Levels of water stress throughout the entire arugula cycle. Soil moisture was normalized by the field capacity of each treatment, thus equaling the references used for measurement. CWSI (Crop Water Stress Index). Treatment 1 (T_1_ = 85% FC (Field Capacity)); Treatment 2 (T_2_ = 75% FC); Treatment 3 (T_3_ = 55% FC). WVC 1 (Volumetric Water Content) for Treatment 1, WVC 2 (Volumetric Water Content) for Treatment 2, WVC 3 (Volumetric Water Content) for Treatment 3.

**Figure 17 sensors-23-01318-f017:**
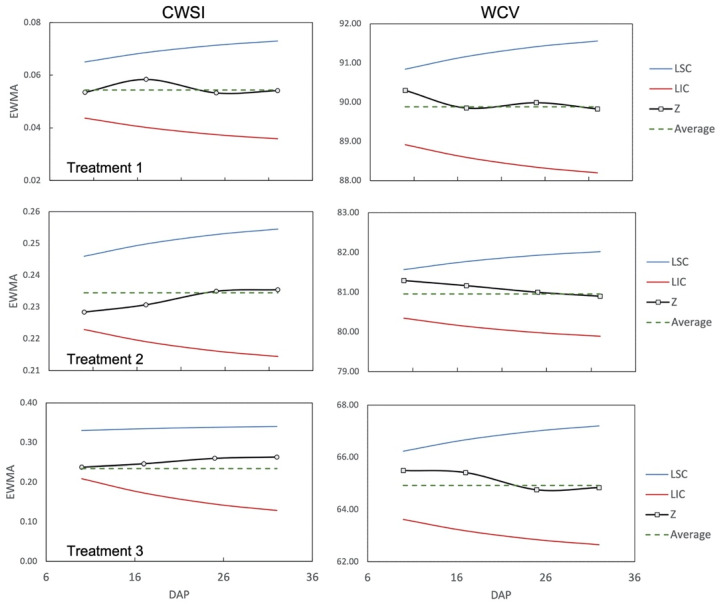
Charts of the exponentially weighted moving average of the CWSI (Crop Water Stress Index) and VWC (Volumetric Water Content) for each treatment. Treatment 1 (T_1_ = 85% FC (Field Capacity)); Treatment 2 (T_2_ = 75% FC); Treatment 3 (T_3_ = 55% FC). LSC (Upper Control Limit), LIC (Lower Control Limit), Z (Exponentially Weighted Moving Average).

**Table 1 sensors-23-01318-t001:** Non-Water-Stressed Baselines (NWSB) obtained in the literature for different crops.

Authors	Crop	NWSB
Idso [7]	Lettuce	Y = −2.96x + 4.18
Erdem [15]	Watermelon	Y = −1.20x + 0.47
Fattahi [16]	Maize	Y = −2.81x − 1.35
Bellvert [28]	Grape	Y = −1.71x + 2.54
Kumar [38]	Mustard	Y = −1.71x − 0.47

**Table 2 sensors-23-01318-t002:** Descriptive statistics and ExG (RGB) evaluation metrics for image processing, where N represents the confidence level (95%).

	Precision	Sensitivity	F-Score	Total Error	Accuracy	Processing Time [s]
Mean	88.79%	84.83%	86.54%	9.96%	90.04%	1.50
Standard error	1.28%	1.27%	0.66%	0.58%	0.57%	0.01
Median	89.96%	85.61%	87.14%	9.99%	90.01%	0.50
Standard deviation	5.10%	5.09%	2.65%	2.31%	2.31%	0.03
Sample variance	0.26%	0.26%	0.07%	0.05%	0.05%	0.00
kurtosis	1.76	3.39	1.56	1.97	1.97	−0.81
Skew	−1.26	−1.55	−0.81	−0.09	0.09	−0.50
Interval	19.04%	21.01%	10.80%	10.47%	10.47%	0.10
Minimum	75.29%	70.53%	80.72%	4.54%	84.99%	0.44
Maximum	94.34%	91.54%	91.52%	15.01%	95.46%	0.54
Score	16	16	15	16	15	15
N	2.72%	2.71%	1.41%	1.23%	1.23%	0.02

**Table 3 sensors-23-01318-t003:** Index values—adapted from Bellvert [28], Cohen [58], Çolak [57], and Mastrorilli [59]. Leaf water potential ($\psi_{L})$, CWSI (Crop Water Stress Index).

Stress Level	ψ*L* [MPa]	CWSI	Reference
Well-irrigated vines	$-0.8\;\leq \psi_{L}\leq -0.6$	CWSI≤0.2	Bellvert [30]
Well-irrigated cotton	$-0.83\;\leq \psi_{L}\leq -1.33$	CWSI≤0.2	Cohen [58]
Well-irrigated quinoa	$-1.25\;\leq \psi_{L}\leq -1.75$	CWSI≤0.25	Çolak [57]
Well-irrigated soybean	$-1.00\;\leq \psi_{L}\leq -1.80$	CWSI≤0.25	Mastrorilli [59]

## Data Availability

Not applicable.

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
