# Peer review of "Water Stress Index Detection Using a Low-Cost Infrared Sensor and Excess Green Image Processing"

_sensors, 2023, doi:10.3390/s23031318_

Round 1

Reviewer 1 Report

Line 236: The reason for applying 0.7 L of water requires explanation. Why not 0.5 or 1.0 L?

Line 285: Specify the stages of the crop.

Table 1: the abbreviation for seconds is sec.

Table 3: check the reference style in the caption.

Figure 25: CVA is mentioned in the caption but WCV is presented in the graph. Correct this mistake.

Author Response

We really appreciate the corrections. We agreed with all of them and implemented them in the new version.

Line 236: The reason for applying 0.7 L of water requires explanation. Why not 0.5 or 1.0 L?

Answer: The reason for being 0.7 l is because the automatic system reservoir has this volume. We clarify this in the text.

Line 285: Specify the stages of the crop.

Answer: Instead of the stages we put the days after planting the images were collected.

Table 1: the abbreviation for seconds is sec.

Answer: Corrected.

Table 3: check the reference style in the caption.

Answer: Corrected.

Figure 25: CVA is mentioned in the caption but WCV is presented in the graph. Correct this mistake.

Answer: Corrected.

Reviewer 2 Report

Interesting research. Study indicates and proves the possibility of obtaining quality information on water stress, using low-cost sensors. Congratulations to the authors.

Author Response

Thank you very much.

Reviewer 3 Report

Reviewer report Manuscript Number: sensors-1956472

Title: Water Stress Index detection using a low-cost Infrared sensor and Excess Green Image processing

Overview comments: This paper proposes a novel computer method combining low-cost thermal sensors and RGB images for computing the CWSI in Eruca sativa M. The idea of proposing a low-cost device for plant water status monitoring is good, having an average novelty because similar studies have been presented in the literature. Unfortunately, the manuscript as it is presented can not be published because it has several form problems and inconsistencies that make it difficult to replicate.

I kindly suggest reading the concerns indicated below.

Abstract:

It must be rewritten after checking the suggestions indicated below in this report. This section should offer relevant information about the problem, research question, methods, main results, and conclusions.

Introduction section:

This section must be improved. It must provide enough support for the research idea, falling from general to particular. I consider that this part of the work was weakly developed. It does not provide enough evidence of previous similar studies or highlight the novelty of the work.  

Lines 36-37: as far as I am aware, the sap flow and stomatal conductance methods are expensive, time-consuming, and challenging to apply for extensive samples. But, I was surprised when I read that the authors indicated: "On the other hand, the stomatal conduction and sap flow methods are not very applicable in the field since they are destructive tests." Please confirm this asseveration.

Line 36: Please confirm the term "stomatal conduction"; it is more frequently used: "stomatal conductance."

Please provide a clear hypothesis that supports this experimental research.

M&M section:

Overview: Please improve this section substantially. This section of the document sounds upside down. It is messy. I suggest explaining in detail the whole experiment and then providing the details about the measurements and data post-processing.

Please indicate whether the experimental FarmBot plot was set up indoors, outdoors, or both.

Please check the references of the whole document. The "Error! Reference source not found" message appears in several parts.

Please add details about the meteorological conditions during the experiment. I suggest checking how the meteorological conditions are presented in similar works; for example, as it was presented in FIg. 2 in: Carrasco-Benavides, M., Antunez-Quilobrán, J., Baffico-Hernández, A., Ávila-Sánchez, C., Ortega-Farías, S., Espinoza, S., Gajardo, J., Mora, M., Fuentes, S., 2020. Performance Assessment of Thermal Infrared Cameras of Different Resolutions to Estimate Tree Water Status from Two Cherry Cultivars: An Alternative to Midday Stem Water Potential and Stomatal Conductance. Sensors 20, 3596. https://doi.org/10.3390/s20123596

Lines 93-94: please add the references

Lines 100-101: please be formal in providing the manufacturer information for this device and all mentioned in the document. I suggest following this order: model, manufacturer, and country of origin, between parenthesis. For example: (A733GSM-GPRS, Adcon Telemetry, Austria).

Line 105: Please add more details about "To."

Lines 116-117: Please indicate the thermal camera resolution.

Lines 114-130: Please add more details about this procedure. The explanation sounds shallow.

Lines 173-187: Please add more details about this procedure. The explanation sounds shallow.

Line 201: Please add references about Eq. 5.

Lines 230-235: Nowhere in the document was it indicated before that a water supply experiment was carried out. I suggest moving this idea to the beginning of the M&M section.

Figures 3 and 4: Please indicate when the indicated Figs are invoked in the paragraphs previous to each one—the same recommendation for the rest of the document's tables and figures.

Fig.3: Please add some flags to identify the different water treatments.

Fig. 4: This Fig. must be moved to the results section. Please add details about all acronyms used in the image. Each image must be self-explanatory. The same for all of

Line 248: whom or what? The soil moisture from Fig 4? Please add more details about the EWMA.

R&D section:

Please improve the discussion of the results substantially. Compare your results against other similar studies. I suggest reviewing the work presented by Osroosh et al., (2018).

Suggested paper:

Osroosh, Y., Khot, L.R., Peters, R.T., 2018. Economical thermal-RGB imaging system for monitoring agricultural crops. Computers and Electronics in Agriculture 147, 34–43. https://doi.org/10.1016/j.compag.2018.02.018

Lines 266-271: Please improve Fig 6 by adding units. Please add a layout of the experiment. It isn't evident when you connect it to Figs 2 and 3.

Line 281-343: Please move this section to M&M.

Line 360: the VPD was not measured; it was computed from temperature and relative humidity. Please correct.

Fig. 11: It is unclear what this image's contribution is. Please improve it. Are the data presented the noontime measurements or the whole day data?

This text is the worst part of this study—the authors present hard-to-follow results and confusing discussions that seriously affect the whole article's quality. The rest of the document, until the end of this section, is chaotic; from Figs 12 ahead, It is unclear how they add relevant information to results and how they connect to the research objectives.

Conclusions section

Please improve it based on the abovementioned comments.

Author Response

Overview comments: This paper proposes a novel computer method combining low-cost thermal sensors and RGB images for computing the CWSI in Eruca sativa M. The idea of proposing a low-cost device for plant water status monitoring is good, having an average novelty because similar studies have been presented in the literature. Unfortunately, the manuscript as it is presented can not be published because it has several form problems and inconsistencies that make it difficult to replicate.

I kindly suggest reading the concerns indicated below.

Answer: We greatly appreciate the corrections and suggestions. We accept all and perform all requested corrections.

Abstract:

It must be rewritten after checking the suggestions indicated below in this report. This section should offer relevant information about the problem, research question, methods, main results, and conclusions.

Answer: Abstract was rewritten.

Introduction section:

This section must be improved. It must provide enough support for the research idea, falling from general to particular. I consider that this part of the work was weakly developed. It does not provide enough evidence of previous similar studies or highlight the novelty of the work.  

Answer: We make a new introduction.

Lines 36-37: as far as I am aware, the sap flow and stomatal conductance methods are expensive, time-consuming, and challenging to apply for extensive samples. But, I was surprised when I read that the authors indicated: "On the other hand, the stomatal conduction and sap flow methods are not very applicable in the field since they are destructive tests." Please confirm this asseveration.

Answer: Thanks for the note, it really was a glitch, we have corrected this inconsistency

Line 36: Please confirm the term "stomatal conduction"; it is more frequently used: "stomatal conductance."

Answer: We change technical term.

Please provide a clear hypothesis that supports this experimental research.

Answer: We make a clear hypothesis.

M&M section:

Overview: Please improve this section substantially. This section of the document sounds upside down. It is messy. I suggest explaining in detail the whole experiment and then providing the details about the measurements and data post-processing.

Answer: Thanks for corrections and advice, we have rewritten this section as per the recommendations.

Please indicate whether the experimental FarmBot plot was set up indoors, outdoors, or both.

Answer: Done.

Please check the references of the whole document. The "Error! Reference source not found" message appears in several parts.

Answer: We used the Mendeley app, for some reason a bug occurred and all references were lost, we redid all references as per journal rule.

Please add details about the meteorological conditions during the experiment. I suggest checking how the meteorological conditions are presented in similar works; for example, as it was presented in FIg. 2 in: Carrasco-Benavides, M., Antunez-Quilobrán, J., Baffico-Hernández, A., Ávila-Sánchez, C., Ortega-Farías, S., Espinoza, S., Gajardo, J., Mora, M., Fuentes, S., 2020. Performance Assessment of Thermal Infrared Cameras of Different Resolutions to Estimate Tree Water Status from Two Cherry Cultivars: An Alternative to Midday Stem Water Potential and Stomatal Conductance. Sensors 20, 3596. https://doi.org/10.3390/s20123596

Answer: Done.

Lines 93-94: please add the references

Answer: Done

Lines 100-101: please be formal in providing the manufacturer information for this device and all mentioned in the document. I suggest following this order: model, manufacturer, and country of origin, between parenthesis. For example: (A733GSM-GPRS, Adcon Telemetry, Austria).

Answer: Done

Line 105: Please add more details about "To."

Answer: Done

Lines 116-117: Please indicate the thermal camera resolution.

Answer: Done

Lines 114-130: Please add more details about this procedure. The explanation sounds shallow.

Answer: Done

Lines 173-187: Please add more details about this procedure. The explanation sounds shallow.

Answer: Done

Line 201: Please add references about Eq. 5.

Answer: Done

Lines 230-235: Nowhere in the document was it indicated before that a water supply experiment was carried out. I suggest moving this idea to the beginning of the M&M section.

Answer: Done

Figures 3 and 4: Please indicate when the indicated Figs are invoked in the paragraphs previous to each one—the same recommendation for the rest of the document's tables and figures.

Answer: Done

Fig.3: Please add some flags to identify the different water treatments.

Answer: Done

Fig. 4: This Fig. must be moved to the results section. Please add details about all acronyms used in the image. Each image must be self-explanatory. The same for all of

Line 248: whom or what? The soil moisture from Fig 4? Please add more details about the EWMA.

Answer: Done

R&D section:

Please improve the discussion of the results substantially. Compare your results against other similar studies. I suggest reviewing the work presented by Osroosh et al., (2018).

Answer: We rewrite the discussion, and cite the work of Oroosh et al. (2018),

Suggested paper:

Osroosh, Y., Khot, L.R., Peters, R.T., 2018. Economical thermal-RGB imaging system for monitoring agricultural crops. Computers and Electronics in Agriculture 147, 34–43. https://doi.org/10.1016/j.compag.2018.02.018

Answer: we used this paper at the end of the discussion of the results.

Lines 266-271: Please improve Fig 6 by adding units. Please add a layout of the experiment. It isn't evident when you connect it to Figs 2 and 3.

Answer: Done

Line 281-343: Please move this section to M&M.

Answer: Done

Line 360: the VPD was not measured; it was computed from temperature and relative humidity. Please correct.

Answer: Done

Fig. 11: It is unclear what this image's contribution is. Please improve it. Are the data presented the noontime measurements or the whole day data?

Answer: Done

This text is the worst part of this study—the authors present hard-to-follow results and confusing discussions that seriously affect the whole article's quality. The rest of the document, until the end of this section, is chaotic; from Figs 12 ahead, It is unclear how they add relevant information to results and how they connect to the research objectives.

Answer: We made an effort to improve this part of the text, making it more assertive and clear.

Conclusions section

Please improve it based on the abovementioned comments.

Answer: Done.

Round 2

Reviewer 3 Report

The document incorporated some of the previous observations, improving it. But, some observations persist, which I invite the authors to check. Please pay attention to the following:

The abstract sound almost similar. It is almost the same. Only some words were changed. Please improve it.

I insist on being formal in providing the manufacturer information for this device and all mentioned in the document. I suggest following this order: model, manufacturer, and country of origin, between parenthesis. For example: (A733GSM-GPRS, Adcon Telemetry, Austria). I made this observation before, but the document has not changed.

Please check each Table and Figure of the document carefully. Each Table and Figure must have the definition of any variable and units used, independent if they were previously mentioned in the manuscript. For example, in Figure 17, who is WCV 1, 2, and 3? In Table 3, what is the meaning of the acronym ψ? ??? The same detail is detected in Fig 18 for mention some. 

Lines 291-316: This text is part of the results of the methodology calibration. The M&M section presents how the researcher experimented, the methods used, and the followed steps, not to mention the results. Thus it should be in the results and discussions section, not the methodology.

Fig. 9 is a mix between Figures and Tables. Fig. 9 does not provide relevant information for the study purposes. It is just an output of statistical software that complements the results from Fig. 8. Thus, I suggest deleting Fig. 9 and just mentioning the results of this analysis and indicating in parentheses "data not shown."

Figure 16 indicates in the caption: "CWSI map generated for 25DAP, using the empirical method to obtain the TMAX and TMIN through natural reference surfaces. One of the leaves was wetted with water to simulate 100% transpiration, and the other was covered with vaseline to block leaf transpiration." How in the M&M section indicated that leaves were painted? Also, the acronyms DAP, TMAX, and TMIN have not been defined in this Figure.

Line 339-345: Why do the authors indicate that? According to line 90. It was an indoor experiment. Thus, how the solar radiation and wind speed influenced the computations? Could the authors clarify it, please?

Please check the references for spelling; for example, the [30] is Bellvet, not Belvet, as mentioned in line 448.

Please improve the discussions; they are almost identical to the previous document. I insist you check similar studies for guidance; please consider this suggestion. You can not discuss your research results on herbaceous plants against results from woody plants, such as the example of vineyards that the manuscript presents. You must search for similar experiences. Previously I suggested using the paper from Osroosh et al., (2018) to follow the discussion this author offered as a model. It was a suggestion to use it as a guide; you can use any other, but please, I ask you to present a good discussion. You can not present a research article with a poor discussion. This part of the study must be improved.

Author Response

The document incorporated some of the previous observations, improving it. But, some observations persist, which I invite the authors to check. Please pay attention to the following:

Answer: We greatly appreciate the corrections and suggestions made. Thanks for the time spent.

The abstract sound almost similar. It is almost the same. Only some words were changed. Please improve it.

Answer: We agree. It was done.

I insist on being formal in providing the manufacturer information for this device and all mentioned in the document. I suggest following this order: model, manufacturer, and country of origin, between parenthesis. For example: (A733GSM-GPRS, Adcon Telemetry, Austria). I made this observation before, but the document has not changed.

Answer: We ask sorry for fail. We did insert.

Please check each Table and Figure of the document carefully. Each Table and Figure must have the definition of any variable and units used, independent if they were previously mentioned in the manuscript. For example, in Figure 17, who is WCV 1, 2, and 3? In Table 3, what is the meaning of the acronym ψ? ??? The same detail is detected in Fig 18 for mention some. 

Answer: Thank you very much for correction. They were done.

Lines 291-316: This text is part of the results of the methodology calibration. The M&M section presents how the researcher experimented, the methods used, and the followed steps, not to mention the results. Thus it should be in the results and discussions section, not the methodology.

Answer: We agree. It was done.

Fig. 9 is a mix between Figures and Tables. Fig. 9 does not provide relevant information for the study purposes. It is just an output of statistical software that complements the results from Fig. 8. Thus, I suggest deleting Fig. 9 and just mentioning the results of this analysis and indicating in parentheses "data not shown."

Answer: We agree. It was done.

Figure 16 indicates in the caption: "CWSI map generated for 25DAP, using the empirical method to obtain the TMAX and TMIN through natural reference surfaces. One of the leaves was wetted with water to simulate 100% transpiration, and the other was covered with vaseline to block leaf transpiration." How in the M&M section indicated that leaves were painted? Also, the acronyms DAP, TMAX, and TMIN have not been defined in this Figure.

Answer: We ask sorry for fail. We describe methods in M&M. Page 6, Line 201. DAP (Day after Planting), TMAXàTUL (Temperature Upper Limit) and TMINàTLL (Temperature Lower Limit) they were defined.

Line 339-345: Why do the authors indicate that? According to line 90. It was an indoor experiment. Thus, how the solar radiation and wind speed influenced the computations? Could the authors clarify it, please?

Answer: Absolutely correct. We corrected

Please check the references for spelling; for example, the [30] is Bellvet, not Belvet, as mentioned in line 448.

Answer: Answer: We ask sorry for fail. We corrected.

Please improve the discussions; they are almost identical to the previous document. I insist you check similar studies for guidance; please consider this suggestion.

Answer: We agree, we found similar studies and we referenced. We insert 11 new papers:

  1. Osroosh, Y.; Hhot, L.R.; Peters, R.T. Econimical thermal-RGB imaging system for monitoring agricultural crops. Computers and Electronics in Agriculture 2018, 147, 34-43, doi.org/10.1016/j.compag.2018.02.018.
  2. Giménez-Gallego, J.; González-Teruel, J.D.; Soto-Valles, F.; Jiménez-Buendía, M; Navarro-Hellín, H. Intelligent thermal image-based sensor for affordable measurement of crop canopy temperature. Agricultural Water Management 2021, 188, doi.org/10.1016/j.compag.2021.106319.
  3. Zhou, Z.; Majeed, Y.; Naranjo, G.D.; Gambacorta, E.M.T. Assessment for crop water stress with infrared thermal imagery in precision agriculture: A review and future prospects for deep learning applications. Computers and Electronics in Agriculture 2021, 182, doi.org/10.1016/j.compag.2021.106019.
  4. Parihar, G.; Saha, S.; Giri, L.I. Application of infrared thermography for irrigation scheduling of horticulture plants. Smart Agricultural Technology 2021, 1, doi.org/10.1016/j.atech.2021.100021.
  5. Luan, Y.; Xu, J.; Lv, Y.; Liu, X.; Wang, H; Liu, S. Improving the performance in crop water deficit diagnosis with canopy temperature spatial distribution information measured by thermal imaging. Agricultural Water Management 2021, 246, doi.org/10.1016/j.agwat.2020.106699.
  6. Katimbo, A.; Rudnick, D.R.; DeJonge, K.C.; Lo, T.H.; Qiao, X; Franz, T.E.; Nakabuye, H.N; Duanm J. Crop water stress index computation approaches and their sensitivity to soil water dynamics. Agricultural Water Management 2022, 266, doi.org/10.1016/j.agwat.2022.107575.
  7. DeJonge, K.C.; Taghvaeian, S.; Trout, T.J.; Comas, L.H. Comparison of canopy temperature-based water stress indices for maize. Agricultural Water Management 2015, 156, 51-62, doi.org/10.1016/j.agwat.2015.03.023.
  8. Çolak, Y.B.; Yazar, A.; Alghory, A.; Tekin, S. Evaluation of crop water stress index and leaf water potential for differentially irrigated quinoa with surface and subsurface drip systems. Irrigation Science 2021, 39, 81-100, doi.org/10.1007/s00271-020-00681-4.
  9. Cohen, V.; Alchanatis, V.; Meron, M.; Saranga, Y.; Tsipris, J. Estimation of leaf water potential by thermal imagery and spatial analysis. Journal of Experimental Botany 2005, 56, 1843-1852, doi:10.1093/jxb/eri174.
  10. Mastrorilli, M.; Katerji, N.; Losavio, N.; Rana, G. Comparison of water stress indicators for soybean. ActaHorticulture 1993, 335.
  11. Bijanzadeh, E.; Moosavi, S.M.; Bahador, F. Quantifying water stress of safflower (Carthamus tinctorius L.) cultivars by crop water stress index under different irrigation regimes. Heliyon 2022, 8, doi.org/10.1016/j.heliyon.2022.e09010.

You can not discuss your research results on herbaceous plants against results from woody plants, such as the example of vineyards that the manuscript presents. You must search for similar experiences.

Answer: We agree, we found a paper for soybean and another for quinoa.

Previously I suggested using the paper from Osroosh et al., (2018) to follow the discussion this author offered as a model. It was a suggestion to use it as a guide; you can use any other, but please, I ask you to present a good discussion.

Answer: We try to improve.

You can not present a research article with a poor discussion. This part of the study must be improved.

Answer: We ask sorry for the failure. We try to improve the discussion.
